



# Empirical evidence for deep convection related stratospheric cirrus clouds over North America

Ling Zou[1,3], Lars Hoffmann[1], Sabine Griessbach[1], Reinhold Spang[2], and Lunche Wang[3]

[1]Jülich Supercomputing Centre, Forschungszentrum Jülich, Jülich, Germany
[2]Institute of Energy and Climate Research (IEK-7), Forschungszentrum Jülich, Jülich, Germany
[3]Hubei Key Laboratory of Critical Zone Evolution, School of Geography and Information Engineering, China University of Geosciences, Wuhan, China

**Correspondence:** Ling Zou (l.zou@fz-juelich.de; cheryl_zou@whu.edu.cn)

**Abstract.** Cirrus clouds in the lowermost stratosphere affect stratospheric water vapor and the Earth's radiation budget. The knowledge of its occurrence and driving forces is limited. To assess the distribution and possible formation mechanisms of stratospheric cirrus clouds (SCCs) over North America, we analyzed SCC occurrence frequencies observed by the Cloud-Aerosol Lidar and Infrared Pathfinder Satellite Observations (CALIPSO) instrument during the years 2006 to 2018. Possible

driving forces such as deep convection are assessed based on Atmospheric Infrared Sounder (AIRS) observations during the same time. Results show that at nighttime, SCCs are most frequently observed during the thunderstorm season over the Great Plains from May to August (MJJA) with a maximum occurrence frequency of 6.2 %. During the months from November to February (NDJF), the highest SCCs occurrence frequencies are 5.5 % over the North-Eastern Pacific, western Canada and 4.4 % over the western North Atlantic. Occurrence frequencies of deep convection from AIRS, which includes storm systems, fronts,

mesoscale convective systems and mesoscale convective complexes at mid- and high latitude, show similar hotspots like the SCCs, with highest occurrence frequencies being observed over the Great Plains in MJJA (4.4 %) and over the North-Eastern Pacific, western Canada and the western North Atlantic in NDJF (~2.5 %). Both, seasonal patterns and daily time series of SCCs and deep convection show a high degree of spatial and temporal relation. Further analysis indicates that the maximum fraction of SCCs related to deep convection is 74 % over the Great Plains in MJJA and about 50 % over the western North

Atlantic, the North-Eastern Pacific and western Canada in NDJF. We conclude that, locally and regionally, deep convection is the leading factor related to occurrence of SCCs over North America. In this study, we also analyzed the impact of gravity waves as another important factor related to the occurrence SCCs, as the Great Plains is a well-known hotspot for stratospheric gravity waves. In the cases where SCCs are not directly linked to deep convection, we found that stratospheric gravity wave observations correlate with SCCs in as much as 30 % of the cases over the Great Plains in MJJA, about 50 % over the North-

Eastern Pacific, western Canada and up to 90 % over eastern Canada and the north-west Atlantic in NDJF. Our results provide better understanding of the physical processes and climate variability related to SCCs and will be of interest for modelers as SCC sources such as deep convection and gravity waves are small-scale processes that are difficult to represent in global general circulation models.





## 1 Introduction

Cirrus clouds have been identified as an important uncertainty component in weather and climate research, with cirrus clouds covering about $20-40\%$ of the globe (Wylie et al., 1994, 2005; Zhou et al., 2014). The wide distribution of cirrus clouds and its radiative properties enables it a crucial factor for understanding the physical composition, dynamic structure and radiative budget of the atmosphere (Liou, 1986). This is especially true for those cirrus clouds in the stratosphere (SCCs), which greatly regulate the water vapor budget in the atmosphere, impact the exchange processes between the troposphere and stratosphere,

and affect the surface energy balance (Roewe and Liou, 1978; Berry and Mace, 2014; Dessler et al., 2016). Although difficult to observe, the presence of SCCs have been found and discussed in a number of studies (Clodman, 1957; Sandhya et al., 2015). However, due to the intricacy of atmospheric processes, precise mechanisms for the formation and existence of SCCs under different atmospheric conditions are not fully understood.

Cirrus clouds in the tropical tropopause layer (TTL) draw a lot of attention as the TTL is the primary entrance of water

vapor from the troposphere to the stratosphere (Boehm and Lee, 2003; Schoeberl et al., 2019). Most tropical cirrus clouds in the lower stratosphere are detected over the western Pacific, South and Southeast Asia, Equatorial Africa, and South America (Massie et al., 2013; Dauhut et al., 2020). However, as shown by Dessler et al. (2013), two-third of the stratospheric water vapor feedback actually was from water vapor increasing in the stratosphere in extratropics. Cirrus clouds in the extratropical lower stratosphere play an important role for the global radiative budget by regulating water vapor in the upper troposphere and

lower stratosphere (Sherwood and Dessler, 2001; Lelieveld et al., 2007). Furthermore, cirrus in the middleworld potentially affects variations of the stratospheric chemical composition, such as the distributions of nitric acid, chlorine and ozone (Borrmann et al., 1996, 1997; Clapp and Anderson, 2019). Therefore, investigating the formation and distribution of extratropical lowermost SCCs is of importance.

Empirical evidence for the existence of SCCs at midlatitudes was found from in situ measurements (Murgatroyd and Gold-

smith, 1956; Clodman, 1957; Bartolome Garcia et al., 2020), ground-based lidar observations (Keckhut et al., 2005; Noël and Haeffelin, 2007) and satellite data (Spang et al., 2015; Homeyer et al., 2017). Cirrus clouds that composed of ice particles, were observed by Nielsen et al. (2007) in the lowermost stratosphere over Southeast Brazil. Sassen and Campbell (2001) showed that about $5-7.5\%$ of cirrus cloud tops are located above the tropopause over Salt Lake City, Utah in the winter season based on a 10-year high cloud data set at the University of Utah Facility for Atmospheric Remote Sensing. About $2.5\%$ of cirrus clouds

with tops above the first thermal tropopause were found over France based on ground-based lidar measurements (Noël and Haeffelin, 2007). Regarding satellite observations, about $5\%$ of the observed cirrus clouds at $40°-60°$ N were detected above the tropopause driven from ERA-Interim reanalysis based on two weeks of infrared limb emission measurements in boreal summer 1997 by the Cryogenic Infrared Spectrometers and Telescopes for the Atmosphere (CRISTA) instrument (Spang et al., 2015). Pan and Munchak (2011) found $\sim2\%$ cirrus cloud with tops being 0.5 km above the tropopause from Cloud-Aerosol Li-

dar and Infrared Pathfinder Satellite Observations (CALIPSO) measurements in both southern and northern midlatitudes based on tropopause heights derived from the National Centers for Environmental Prediction Global Forecast System (GFS), while about $4\%$ stratospheric cirrus cloud with tops being 0.5 km above the ERA-Interim reanalysis' first tropopause over latitudes





between $40° – 60°$ were estimated from Michelson Interferometer for Passive Atmospheric Sounding (MIPAS) observations (Zou et al., 2020). Even though results are kind of controversial from study to study, the occurrence of SCCs at midlatitudes is notable.

Considering the pathways of air into the midlatitude stratosphere, deep convection, especially tropopause-penetrating convection, could produce direct injection of air into the middleworld and even the overworld stratosphere (Weinstock et al., 2007). Smith et al. (2017) observed amount of water vapor with depth of $\sim 2\,\mathrm{km}$ in the stratospheric overworld ($380\,\mathrm{K}$ to $415\,\mathrm{K}$) in the south of the Great Lakes on 27 August 2013, then they found that the detrainment and mixing of water in the stratosphere was likely traced back to the tropopause penetrating storm system. On the base of the National Oceanic and Atmospheric Administration Next Generation Weather Radar (NEXRAD) network over the continental United States, Cooney et al. (2018) and Solomon et al. (2016) found that the overshooting events are most common in the central of the United States. Cooney et al. (2018) discovered about $45\,\%$ of the overshooting events with echo tops at least 1 km above the tropopause extended above the 380 K level into the stratosphere. Yu et al. (2020) also found $\sim 1$ ppmv moister air in the stratosphere in summer time when deep convection occurs over the North American region based on Microwave Limb Sounder data. Deep convection contributes to the water budget in the midlatitude stratosphere. Also as convective cores may produce the detrainment and mixing of water, predominantly in the form of ice (Smith et al., 2017), the cirrus clouds in the stratosphere would be linked to the deep convection.

Gravity wave generated temperature perturbations are important for the formation of ice clouds (Haag and Kärcher, 2004; Jensen and Pfister, 2004). Kim et al. (2016) and Podglajen et al. (2018) observed the formation and existence of high altitude cirrus clouds in cold phase of upward propagating waves in the TTL based on the Airborne Tropical TRopopause EXperiment (ATTREX) measurements. In midlatitudes, the formation of cirrus cloud caused by mesoscale gravity waves was found and simulated by Kärcher and Podglajen (2019). For gravity wave induced ice clouds in the stratosphere, Hoffmann et al. (2017) investigated the gravity waves from the Atmospheric Infrared Sounder (AIRS) observations, and found the temperature fluctuations yielded by gravity waves triggered the formation of polar stratospheric clouds. The turbulence and mixing generated by breaking gravity waves would influence the occurrence of cirrus clouds in the lower stratosphere. Homeyer et al. (2017) found gravity wave breaking was the primary source for above-anvil cirrus plumes, which could reach up to $1 – 6\,\mathrm{km}$ above the tropopause in extratropical convective system. According to Wang (2003), turbulent mixing and advection downwind at the top of overshooting were potentially contributing to the formation of cirrus plumes above anvil, in which gravity wave breaking sets up strong vertical motions at the cloud top, injects the water vapor into the stratosphere and leads to cirrus plumes above the anvils. The relation between gravity waves and the formation and occurrence of stratospheric cirrus clouds is noteworthy.

Moreover, poleward transport of water vapor from the TTL to the extratropics is an important factor for cirrus clouds formation (Dessler et al., 1995; Pittman et al., 2007). Spang et al. (2015) found a significant amount of cirrus clouds in the lowermost stratosphere at mid- and high latitudes based on CRISTA satellite observations, and it turned out that poleward isentropic transport from the upper tropical troposphere to the extratropical stratosphere lead to the occurrence of SCCs in these observations. Deep convection associated with double tropopause events contributed to the presence of stratospheric cirrus at altitudes of $1 – 2\,\mathrm{km}$ above the tropopause (Homeyer et al., 2014b). Petzold et al. (2020) found potential relations between cirrus and the North





Atlantic Oscillation (NAO) index in the extratropical upper troposphere and lowermost stratosphere at northern midlatitudes over the eastern North American, North Atlantic and European regions for the time period from 1995 to 2010. Furthermore,

freezing aerosol particles in the upper troposphere and lower stratosphere (UTLS) might also influence the formation of sub-visible cirrus at midlatitudes (Gierens et al., 2000). Above all, the mechanisms leading to the formation of cirrus clouds are manifold and complicated. Specifying the relations of driving factors and SCCs is crucial for better understanding the potential effects of SCCs in different regions and seasons.

Considering the relevance of SCCs regarding their impact on the Earth's radiation budget and climate change, we analyzed

a nearly 13-year record (2006 to 2018) of satellite observations of SCCs and deep convection over North America. The study area of North America was mainly selected to enable comparison of the results with other studies focusing on that region. The main objectives of our work are i) to quantify the occurrence and distribution of SCCs over North America, ii) to quantify the contribution of deep convection as a primary force for the formation of SCCs in that region and iii) to discuss potential other driving factors related to the presence of SCCs, in particular gravity wave events and atmospheric transport.

Further information on the CALIPSO and AIRS instruments, the ERA-Interim tropopause data and the detection methods for SCCs, deep convection and gravity waves is presented in Section 2. In Section 3, we discuss occurrence frequencies of SCCs, deep convection, gravity waves and their relationships, as well as the contribution of deep convection and gravity waves on the formation of SCCs over the North America. In Section 4, we further discuss the role of deep convection, gravity waves and transport from deep convection in the formation and maintenance of SCCs. Summary and conclusions are presented in

Section 5.

## 2   Data and methods

### 2.1   Tropopause data

The subdivision of different atmospheric layers is often based on the vertical temperature structure of the atmosphere. The lapse rate tropopause (LRT) is the lowest level at which the lapse rate decreases to $2°$ C/km or less, provided the average

lapse rate between this level and all higher levels within 2 km does not exceed $2°$ C/km (WMO, 1957). The LRT is a globally applicable tropopause definition to identify the transition between the troposphere and stratosphere (Munchak and Pan, 2014; Spang et al., 2015; Xian and Homeyer, 2019). Here, the ERA-Interim reanalysis (Dee et al., 2011) has been used to derive LRT geopotential heights (Hoffmann, 2020a). ERA-Interim is a global meteorological reanalysis with approximately $0.75°$ horizontal grid resolution on 60 vertical levels from the surface up to 0.1 hPa, which is available 6-hourly from 1979 to August

2019. The vertical resolution of the data is about 600-1000 m for tropopause heights ranging from 9 to 18 km. However, the ERA-Interim data have been interpolated to a much finer vertical grid using cubic spline interpolation in order to improve the vertical resolution of the geopotential height estimates of the tropopause.



## 2.2 CALIPSO observations of SCCs

The Cloud-Aerosol Lidar with Orthogonal Polarization (CALIOP) is a near-nadir viewing two-wavelength (532 nm and 1064 nm)
polarization-sensitive lidar instrument (Winker et al., 2007, 2009), which is carried by the Cloud-Aerosol Lidar and Infrared
Pathfinder Satellite Observations (CALIPSO) satellite. CALIOP probes high-resolution vertical structures and properties of
clouds and aerosols on a nearly global scale. CALIPSO was launched on 28 April 2006 and remained part of NASA's 'A-
Train' satellite constellation until September 2018. The A-Train satellite constellation operates at a nominal altitude of 705 km
with 98° inclination and has equatorial crossing times at 01:30 and 13:30 local time (LT).

The latest released CALIPSO version 4.x (V4.x) data products provide significantly improved Level-2 data by using a new
cloud-aerosol discrimination algorithm and improved extinction retrieval algorithms (Liu et al., 2019; Young et al., 2018).
Those improved algorithms provide more reliable identification of aerosol and cloud layers and higher lidar ratios for ice
clouds, which make the data product more suitable for studying cirrus cloud. To reliably estimate cirrus clouds top heights,
only high-feature-type-quality cirrus cloud data from the CALIPSO V4.x Level-2 Vertical Feature Mask data product are used
in this study. Day- and nighttime data need to be analyzed separately due to their different signal-to-noise ratios and detection
sensitivity (Getzewich et al., 2018).

Factoring in the uncertainties of the tropopause and cloud top heights, we identify cirrus clouds with top heights of more than
500 m above the ERA-Interim LRT tropopause as SCCs in this study. The same approach to identify SCCs from CALIPSO
observations was already used by Zou et al. (2020). The 500 m threshold for SCC detection is also comparable to the approach
of Homeyer et al. (2010) and Pan and Munchak (2011). Considering that cirrus layers near the tropopause can extend horizon-
tally over hundreds to thousand kilometers (Winker and Trepte, 1998), a grid size of $4° × 6°$ (latitude $×$ longitude), which may
involve those large scale SCCs and deep convection systems and also provide less noisy signals in figures, is used in our study
to investigate occurrence frequencies of SCCs.

## 2.3 AIRS observations of deep convection and gravity waves

The Atmospheric Infrared Sounder (AIRS) (Aumann et al., 2003; Chahine et al., 2006) is one of six instruments aboard
NASA's Aqua satellite. Being the first satellite in the A-Train constellation, Aqua also has a nominal orbital altitude of 705 km
and inclination of 98°. It shares the same equatorial crossing time with other A-Train members for the descending (01:30 LT)
and ascending (13:30 LT) orbit sections. AIRS measures the thermal emissions of atmospheric constituents in the nadir and
sublimb viewing geometry. The AIRS infrared spectrometer has a total of 2378 spectral channels with a spectral coverage of
3.74 to 4.61 $\mu$m, 6.20 to 8.22 $\mu$m and 8.8 to 15.4 $\mu$m. Over the full dynamic range from 190 K to 325 K, the absolute accuracy
of each spectral channel is better than 3 % and noise is less than 0.2 K at 250 K scene temperature (Aumann et al., 2000). The
AIRS Level-1B (version 5.x) radiance data have been used to extract information of deep convection (Aumann et al., 2006;
Hoffmann and Alexander, 2010) and gravity waves (Hoffmann et al., 2013, 2014) in this study.

As optically thick clouds at upper altitudes show lower brightness temperatures than clouds at lower altitudes or the surface
at clear sky conditions, particularly low brightness temperatures in spectral window regions can be used to detect deep convec-





tion and strong storm systems from infrared nadir sounder observations. To detect deep convection at latitudes below $\pm\,60°$ from AIRS measurements, Aumann et al. (2003) used a brightness temperature threshold of 210 K for the 1231 cm$^{-1}$ (8.1 $\mu$m) radiance channel. Hoffmann and Alexander (2010) increased the threshold to 220 K to better identify convective systems at midlatitudes, which appear at higher tropopause temperatures at these latitudes. Later, Hoffmann et al. (2013) pointed out that

ambiguous detections at different latitudes and seasons are likely caused by using a constant threshold for detection. Therefore, varying thresholds, which are based on monthly and latitudinally mean tropopause temperatures from the NCAR/NCEP reanalysis, have been used in their work. Following this approach, we use co-located tropopause temperatures from the ERA-Interim reanalysis to achieve further improved identification of deep convective events. By considering the vertical grid spacing of the ERA-Interim reanalysis and a sensitivity test (Appendix A), an offset of $+7$ K on top of the tropopause temperature ($T_{TP}$)

was selected as a threshold to detect deep convection events from AIRS brightness temperature measurements (BT$_{AIRS}$) at 1231 cm$^{-1}$ here,

$$BT_{AIRS} - T_{TP} <= 7\,K. \tag{1}$$

Note that the term 'deep convection' used in this work does not only refer to convection from tropical storms, but also to strong convective events from various sources such as storm systems and fronts, mesoscale convective systems and mesoscale

convective complexes at mid- and high latitudes. Sensitivity tests on deep convection detection can be found in Appendix A.

Mean brightness temperatures in the carbon dioxide 4.3 $\mu$m waveband have previously been used to detect stratospheric gravity wave signals from AIRS observations (e.g., Hoffmann and Alexander, 2010; Hoffmann et al., 2013; Yue et al., 2013; Hoffmann et al., 2018). In order to reduce noise and to improve the detection sensitivity, we averaged measurements of 42 AIRS channels from 2322.6 to 2345.9 cm$^{-1}$ and 2352.5 to 2366.9 cm$^{-1}$. In the next step, background signals from planetary

waves and large-scale temperature gradients are removed by means of a low pass filter. To identify gravity wave events, a variance filter was then applied to the 4.3 $\mu$m brightness temperature perturbations. Gravity wave events are detected based on a threshold of 0.05 K$^2$ for the noise-corrected 4.3 $\mu$ brightness temperature variances. The detection method described here is sensitive to a broad range of gravity wave horizontal wavelengths (50 to 1000 km), but limited to vertical wavelengths of about 15 km or longer. The method can be used to detect gravity waves in the middle and upper stratosphere ($30-40$ km of altitude)

(Hoffmann and Alexander, 2010; Hoffmann et al., 2013, 2018). However, as gravity waves typically propagate upward from their tropospheric sources into the stratosphere, the detections also provide information about gravity waves at tropopause levels.

## 3 Results

### 3.1 Occurrence frequencies of SCCs and deep convection

Figure 1 presents a nighttime observation of SCCs above a thunderstorm on 12 June 2018 over the North American Great Plains. Brightness temperatures as low as $\sim$192 K indicate low cloud top temperatures related to deep convection and red dots show SCCs along the CALIPSO orbits (Fig. 1, upper panel). The corresponding 532 nm total attenuated backscatter (bottom

panel) indicates that the top of the convection overshoots the tropopause into the lower stratosphere by up to ∼2.7 km and cirrus clouds are present at the top of the convective center and above anvils. The observations of deep convection by AIRS

and SCCs by CALIOP show a high degree of consistency regarding occurrence time and location.

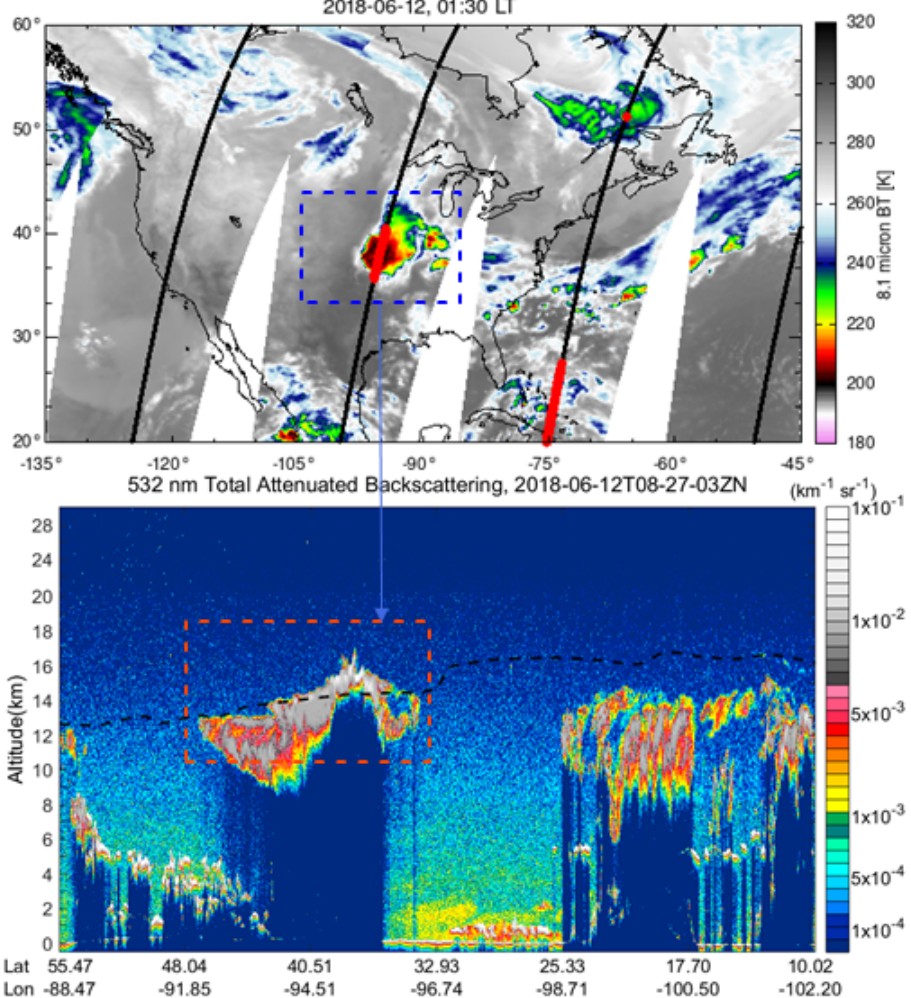

**Figure 1.** SCCs observed above deep convection on 12 June 2018. a) 8.1 μm brightness temperatures (BT) derived from AIRS overlayed with CALIPSO orbits (black lines). The red circles along the orbits indicate cirrus with cloud top heights being at least 500 m above the tropopause. b) shows total attenuated backscatter for a whole CALIPSO orbit from a), the region of interest lies within the red dotted region only. The dashed black line indicates the tropopause height derived from ERA-Interim.

This example of SCCs co-located with deep convection shown here is rather typical, and cases like this are frequently observed in the CALIPSO and AIRS data during the thunderstorm season over the North American Great Plains. The example adds a further aspect that deep convection would contribute to the formation of SCCs to previous studies showing the significant contribution of convection on the water vapor distribution in the lower stratosphere over North America (e.g., Ray et al., 2004;





Sun and Huang, 2015; Qu et al., 2020). Investigating the relationship between SCCs and deep convection will provide further insights into the potential mechanism of SCC formation and water vapour transport into the stratosphere over North America.

The occurrence frequencies of SCCs and deep convection for the summer season, i. e., May to August (MJJA), and the winter season, i. e., November to February (NDJF), were calculated from CALIPSO measurements between 2006 and 2018 as the ratio of the number of detections to the total number of profiles in $4° \times 6°$ (latitude $\times$ longitude) grid boxes (Fig. 2). As

the CALIPSO measurements have a better signal-to-noise ratio at nighttime than at daytime (Winker et al., 2009; Hunt et al., 2009), which has a significant impact on the detection of SCCs (Pan et al., 2009; Zou et al., 2020), we focus on the analysis of nighttime data in this study.

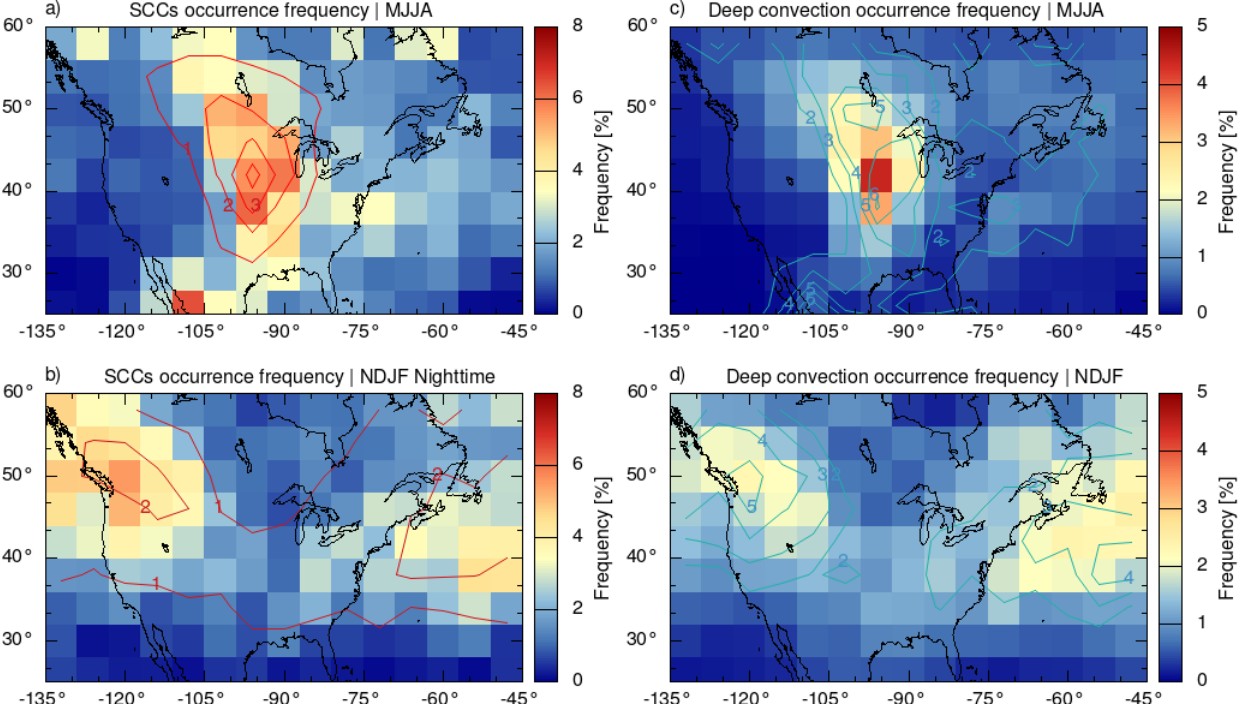

**Figure 2.** Nighttime seasonal mean occurrence frequencies of SCCs from 2006 to 2018 in a) MJJA and b) NDJF on a $4° \times 6°$ (latitude $\times$ longitude) grid size. SCCs detected by CALIPSO are cirrus clouds that have a cloud top height being at least 500 m above the tropopause derived from ERA-Interim. Corresponding occurrence frequencies of deep convection for c) MJJA and d) NDJF. Deep convection events measured by AIRS are defined as a difference of the 8.1 $\mu$m brightness temperature and the ERA-Interim tropopause temperature below 7 K. The red contour lines in a) and b) show the occurrence frequencies of deep convection and blue contour lines in c) and d) are occurrence frequencies of SCCs.

Over North America and the neighbouring ocean regions, SCCs are most often detected over the Great Plains during the Midwest thunderstorm season (MJJA) with maximum occurrence frequencies of 6.2 % (Fig. 2a) and in the winter season (NDJF)

high SCC occurrence frequencies are found over the North-Eastern Pacific, western Canada (up to 5.5 %) and the western North Atlantic with maximum occurrence frequencies of 4.4 % (Fig. 2b). These occurrence frequencies are in agreement with





results reported for a shorter time span (2006 to 2012) of CALIPSO measurements (Zou et al., 2020), in which the SCCs were retrieved with the same detection method and tropopause data.

The patterns of the occurrence frequencies of deep convection from AIRS, which were calculated as the ratio between the
number of deep convection pixels and the total number of pixels in each grid box, are quite similar to the CALIPSO SCC observations (Fig. 2). High occurrence frequencies of deep convection occur over the Great Plains in MJJA with maximum value of 4.4 % (Fig. 2c) and over the North-Eastern Pacific, western Canada and the western North Atlantic in NDJF reaching 2.5 % (Fig. 2d). The patterns and absolute values of nighttime occurrence frequencies of deep convection found here are in line with results of Hoffmann and Alexander (2010), Hoffmann et al. (2013) and de Groot-Hedlin et al. (2017) using AIRS data for
different ranges of years.

Even though at midlatitudes the SCC occurrence maxima are slightly shifted downwind (eastward) compared to the deep convection maxima, the occurrence frequencies overall indicate a high degree of relation between SCCs and deep convection.

### 3.2   Temporal correlations of SCCs and deep convection

The results from Sect. 3.1 show that the hotspots of SCCs and deep convection are spatially overlapping over the Great Plains
in MJJA and over the North-Eastern Pacific, western Canada and western North Atlantic in NDJF. To further analyse the spatial relations, yearly nighttime mean occurrence frequencies of SCCs and deep convection in MJJA from 2007 to 2018 are presented in Fig. 3. This analysis shows that hotspots of SCCs and deep convection are often simultaneously detected over the Great Plains, even though the magnitude of the occurrence frequencies of SCCs and deep convection change significantly from year to year. Maximum occurrence frequency of SCCs reach up to 15 % in 2011 with the largest occurrence frequency of deep
convection of more than 6 %. In the years 2012, 2016 and 2017, occurrence frequencies of SCCs are as low as $3-6$ %, with weaker frequencies of deep convection of $3-4$ %. In all years, the peak frequencies of SCCs usually appear around the center of deep convection.

The temporal correlations between SCCs and deep convection are further analyzed with time-series of daily detection numbers ($n_{obs}$) over the Midwest United States ($35°N-45°N$, $90°W-100°W$) providing the number of days with detections of
SCCs ($NOD_{SCC}$), number of days with detections of deep convection ($NOD_{DC}$), the minimum brightness temperature from AIRS ($BT_{min}$) and the Pearson linear correlation coefficients of SCCs and deep convection ($r_{SCC-DC}$), SCCs and minimum brightness temperature ($r_{SCC-BTmin}$) for the summer season (MJJA) of each year (Fig. 4 and Fig. 5). Despite large day-to-day variations, the occurrences of SCCs and deep convection are generally correlated. The linear correlations between SCCs and deep convection are particularly high in 2007, 2010 to 2012 and 2017 with correlation coefficients being larger than 0.6.
The days with SCCs are mostly accompanied by deep convection, even in 2015, which shows the lowest correlation coefficient of about 0.3. The daily minimum brightness temperature in interested region, which may provide clues for the intensity of deep convection, are all negatively correlated with SCCs in each year.

Note that the numbers of SCC and deep convection events are not necessarily linearly correlated. The rate at which SCCs are formed by deep convection will depend on several parameters such as the intensity and the spatial extent at which deep
convection occurs. The brightness temperature threshold method applied to the AIRS observations to detect deep convection





captures this complexity only to some extent. Rather than a linear correlation analysis on the number of detections, a more simple analysis is to look at the relations between the number of days with SCC and deep convection occurrences. This analysis shows that more than 80 % of the days with SCCs co-occur with days with deep convection. The days with SCCs are generally co-occurring with lower brightness temperatures which indicates the intensity of deep convection would affect the

occurrence frequencies of SCCs. In summary, the analyses show a high degree of correlation on the temporal scale between deep convection and SCCs over the Great Plains in summertime.



**Figure 3.** Nighttime mean occurrence frequencies of SCCs from CALIPSO (shown as colored boxes) and deep convection from AIRS (shown as red contour lines) in MJJA from 2007 to 2018.





**Figure 4.** Daily numbers of nighttime observations of SCCs (orange line with no offset), deep convection (green line with offset of 300) and minimum brightness temperature (purple line) over the Midwest United States ($35°N - 45°N$, $90°W - 100°W$) in MJJA from 2007 to 2012. Number of days with occurrences of SCCs ($NOD_{SCC}$), deep convection ($NOD_{DC}$) and both of them ($NOD_{both}$) are counted. Total numbers of SCCs ($n_{SCC}$), deep convection ($n_{DC}$) and the minimum brightness temperature ($BT_{min}$) in MJJA and their Pearson linear correlation coefficients ($r_{SCC-DC}$ and $r_{SCC-BTmin}$) are also shown for each year.



**Figure 5.** Continued from Fig. 4, but for the years from 2013 to 2018.





### 3.3 SCCs related to deep convection

In this section, the relations between SCCs and deep convection are investigated in more detail by analyzing the fraction of SCC observations occurring in the same grid box with deep convection on the same day at nearly the same time, considering that both AIRS and CALIPSO are in the A-Train (Fig. 6). The fraction of SCCs related to deep convection is defined as the ratio of day numbers with SCCs and deep convection both detected to the total number of days with SCC detections. In MJJA, most SCC observations are directly related to deep convection over the Great Plains with the highest fraction up to 74 % (Fig. 6a). In NDJF, the SCCs observations over the western Atlantic Ocean (up to 50 %) and secondly over the North-Eastern Pacific and western Canada with maximum fraction of 44 % are mostly related to deep convection. Regions with a high degree of relation between SCCs and deep convection agree best with the regions with large occurrence frequencies of deep convection, indicating that deep convection is the main factor related to the formation of SCCs in the study area.

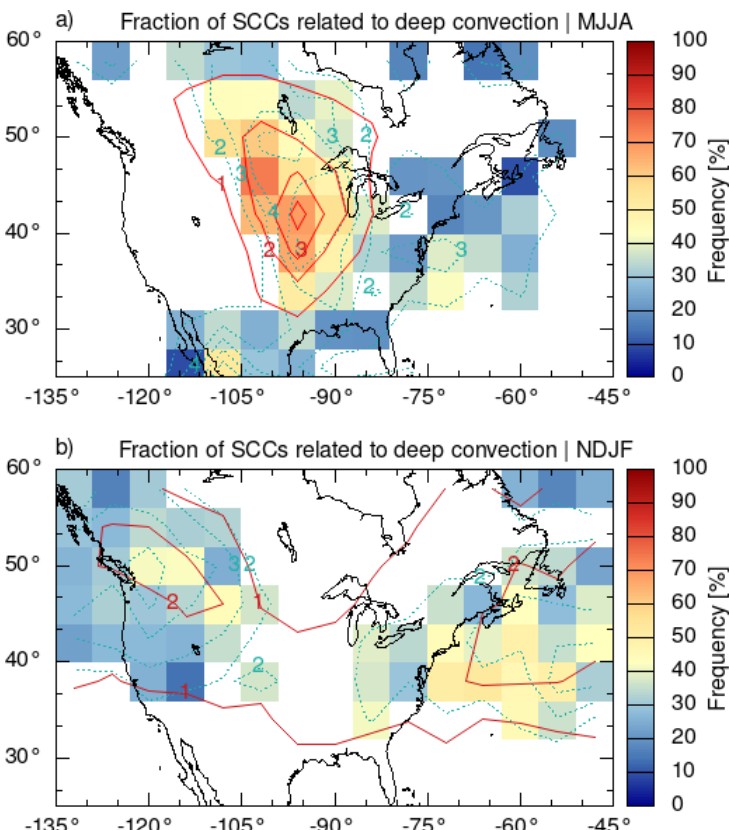

**Figure 6.** Colored boxes indicate the fraction of SCCs related to deep convection. Blank grid cells are regions with SCC occurrence frequencies less than 2 %. Regions with SCC occurrence frequencies of >=2 % are shown with dotted blue contour lines and regions with deep convection occurrence frequencies of >=1 % are shown with red contour lines.





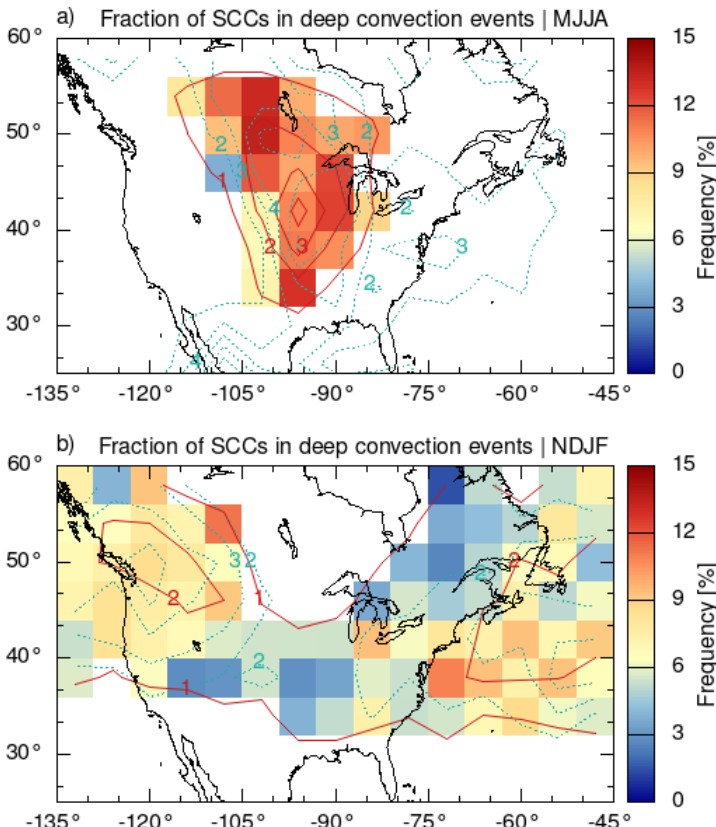

**Figure 7.** Fraction of deep convection events related to SCCs. The red contour lines indicate the occurrence frequencies of deep convection and blue dotted contour lines show the occurrence frequencies of SCCs.

While the majority of the SCCs is directly related to deep convection in the investigation area, only about 12 % of the deep convection events co-occur on average with SCCs over the Great Plains in MJJA and less than 10 % over the North-Eastern Pacific, western Canada and the western North Atlantic in NDJF (Fig. 7).

After removing the SCC observations directly related to deep convection, SCC occurrence frequencies as large as 3 % over the Great Plains in MJJA (Fig. 8a) and 3.6 % over the North-Eastern Pacific, western Canada and 3 % over the western North Atlantic in NDJF (Fig. 8b) remain. In summer (MJJA) the maximum is located to the east of the deep convection hotspot over the Great Plains, and also, to a lesser extent the NDJF maximum is located to the east of the convection hotspot, The downwind shift of SCCs to deep convection is apparent, especially in summer time. This indicates other mechanisms such as atmospheric

transport need to be further investigated.



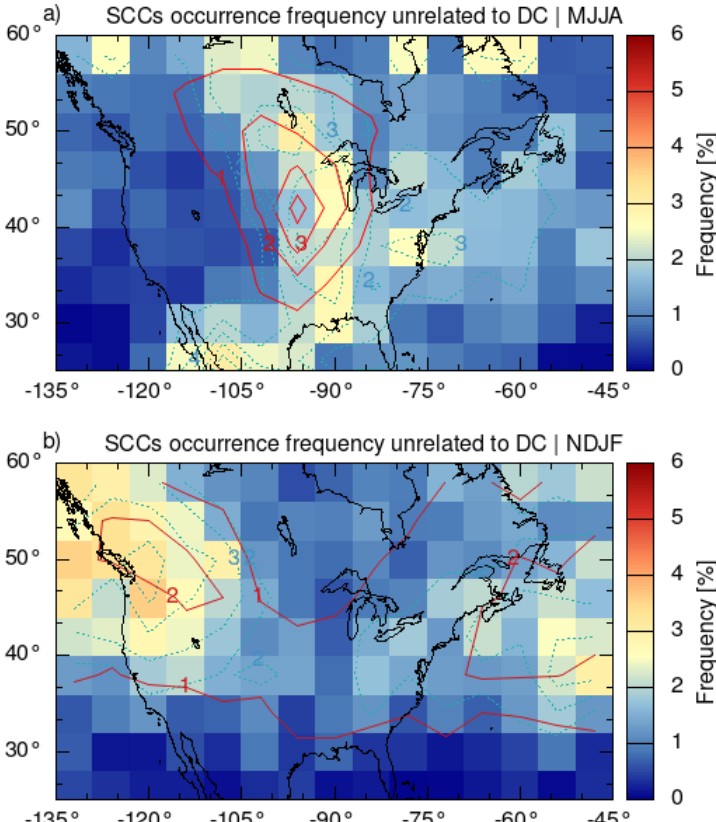

**Figure 8.** Occurrence frequencies of SCC observations that are not directly related to deep convection. The red contour lines show the occurrence frequencies of deep convection and blue dotted contour lines are the occurrence frequencies of SCCs.

## 3.4 SCCs and atmospheric transport

A group of SCCs detected on 12 May 2011 over the Great Plains, which is unrelated to local deep convection, is shown in Fig. 9a. To trace the potential origin of those SCCs, the highly scalable Massive-Parallel Trajectory Calculations (MPTRAC) model driven by ERA5 reanalyses (Hoffmann et al., 2016, 2019; Hoffmann, 2020b) was used to calculate 3 days' backward

trajectories. The red triangles in Figs. 9c and 9d indicate the location of the SCC observation and blue triangles mark the position of the air parcels at local time 13:30 and 01:30 on 11 May 2011. Convection systems are observed along the trajectory by AIRS and are possibly the origin of the SCC. In this case, the SCCs formed from the convective system and were transported to the northeast direction by the jet stream.

The example further supports the finding in Sect. 3.3 that SCCs are also observed downwind of deep convection regions.
This may in particular explain some of the high occurrence frequencies of SCCs over the Great Plains in Fig. 8, as they are located downwind of a deep convection hotspot. To enable a more detailed atmospheric transport analysis, a practical method by simply enlarging the search radius for deep convection in the grid box is applied in our work, which enlarges the search




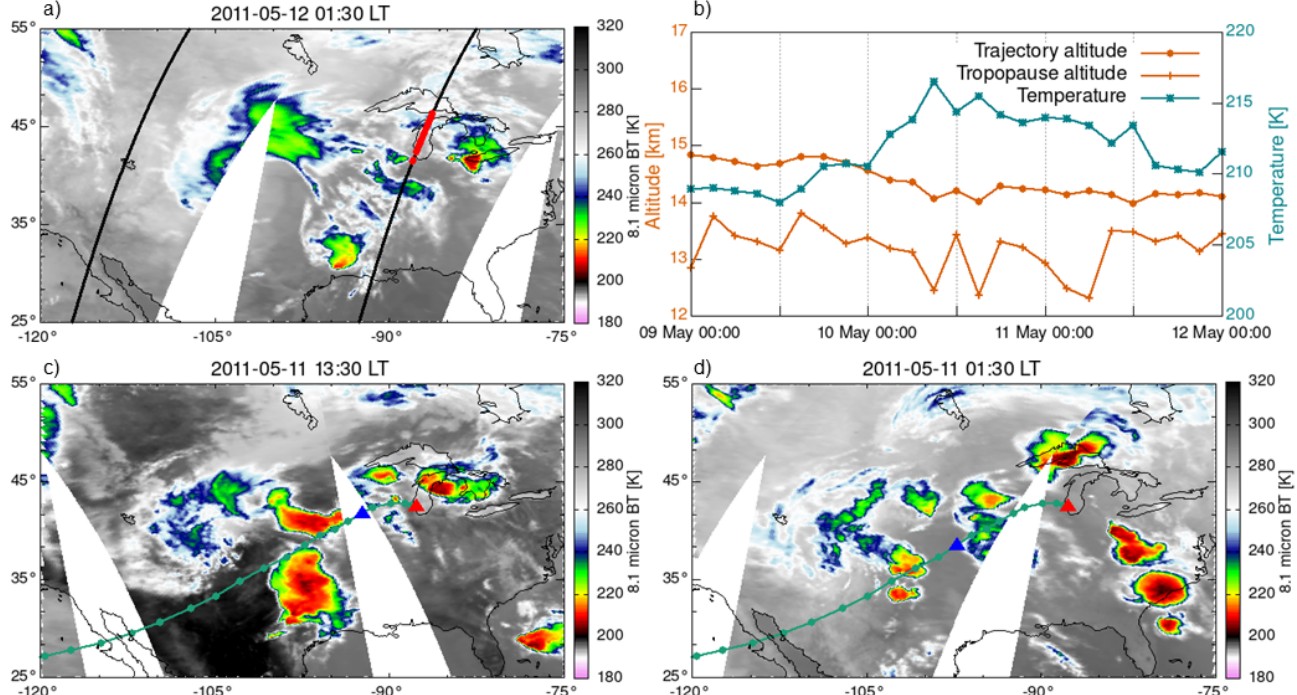

**Figure 9.** CALIPSO SCC (red dots) and AIRS deep convection (colours) observations. a) The observed SCCs on 12 May 2011 are unrelated to local deep convection. b) The trajectory height, tropopause height and trajectory temperature along a three days backward trajectory. c) and d) The trajectory (green dotted line with a time interval of three hours) starting at an SCC observation (red triangle) passes deep convection systems approximately several hours earlier on 11 May 2011. The blue triangles indicate the position of the air parcel at local time 13:30 and 01:30 on 11 May 2011.

radius from $4° \times 6°$ to $6° \times 8°$ (latitude $\times$ longitude). The fractions of SCCs related to a larger search radius' deep convection are presented in Fig. 10. Although the patterns are very similar, the fractions of SCCs related to deep convection now increase
up to 83 % over the Great Plains in MJJA, 55 % over the North-Eastern Pacific and western Canada and 62 % over the western North Atlantic in NDJF. These values are generally 10 % larger than the results shown in Fig. 6. This suggests that transport from deep convective sources is another important factor for the occurrence of SCCs in the study area.



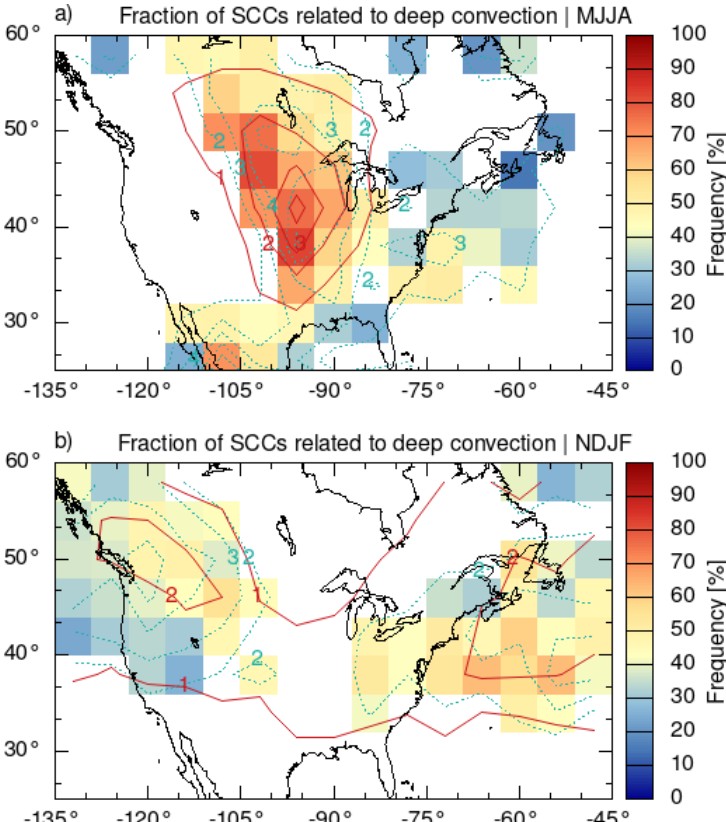

**Figure 10.** Fraction of SCCs related to deep convection with a larger search radius of $6° \times 8°$ (latitude $\times$ longitude) for deep convection detection. The red contour lines are the occurrence frequencies of deep convection and blue dotted contour lines are the occurrence frequencies of SCCs.

### 3.5 SCCs related to gravity waves

The summertime deep convection over the Great Plains is known to induce gravity waves (Hoffmann and Alexander, 2010) and
gravity waves are known to contribute to cirrus formation (Jensen et al., 2016). To address potential sources for the SCCs not directly related to deep convection as found in Fig. 8, we analyzed the SCC relation with gravity waves. An example of SCCs induced by gravity waves is shown in Fig. 11. The top of the SCCs is located about $1.2\,\mathrm{km}$ above the tropopause according to the CALIPSO observations. The AIRS $4.3\,\mu$m brightness temperature variances around the SCCs are about $0.06\,\mathrm{K}^2$, which exceeds the significance level of AIRS to detect stratospheric gravity waves. The stratospheric gravity waves were likely caused
by a convective system to the west of the SCCs as indicated by the propagation direction of the waves patterns (Fig. 11).

Counting only SCCs connected with gravity waves in absence of deep convection in the same grid box, Fig. 12 shows the occurrence frequencies of gravity waves as well as the fraction of SCCs related to gravity waves. In contrast to deep convection, gravity waves are more often detected in NDJF over eastern Canada and the north-west Atlantic ($>30\,\%$) (Fig. 12b). In MJJA,



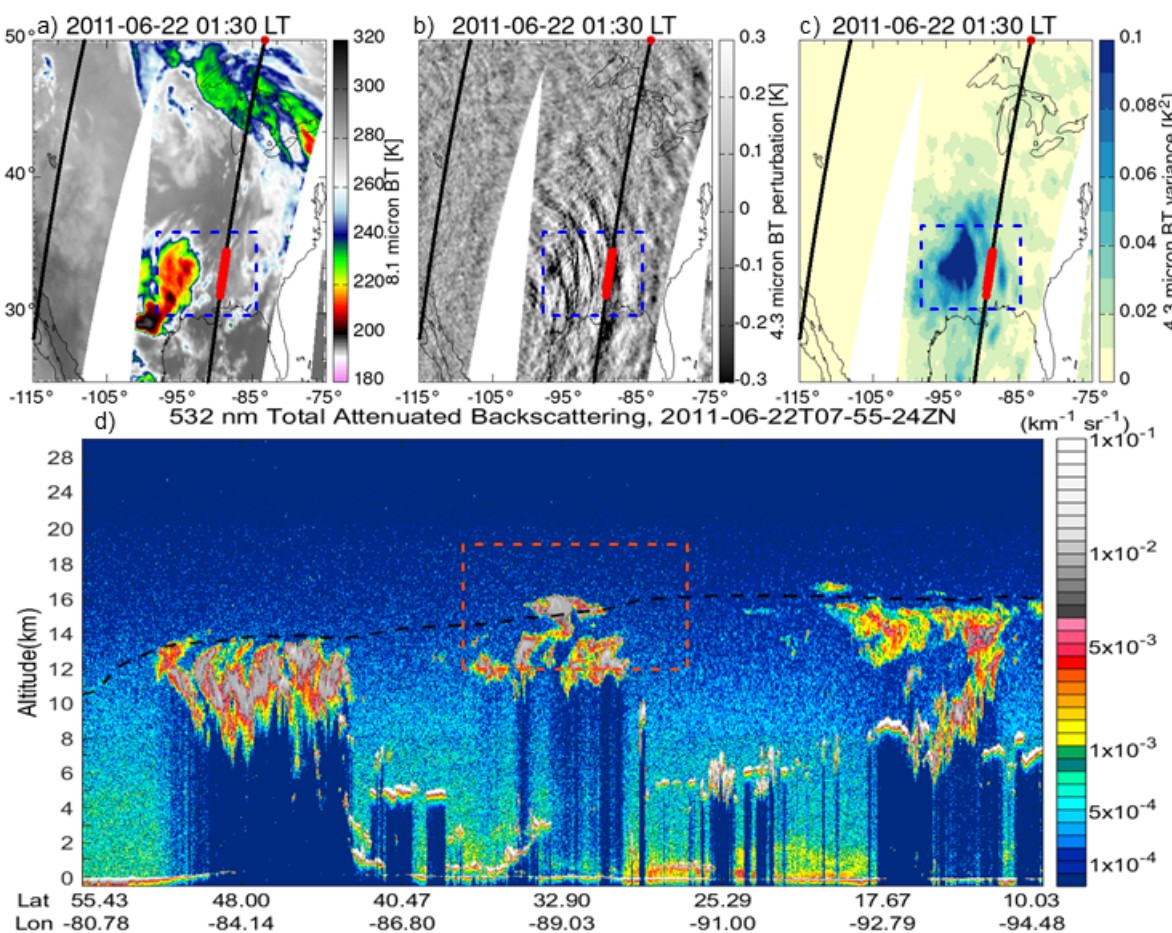

**Figure 11.** Example of relations between gravity waves and SCCs. Upper panels show brightness temperature, brightness temperature perturbations and variances from AIRS measurements, brightness temperature variances in c) with threshold of $0.05\ \text{K}^2$ identify stratospheric gravity waves. Red dots indicate SCC detections along CALIPSO orbits. The bottom panel shows vertical features of clouds from CALIPSO backscatter data.





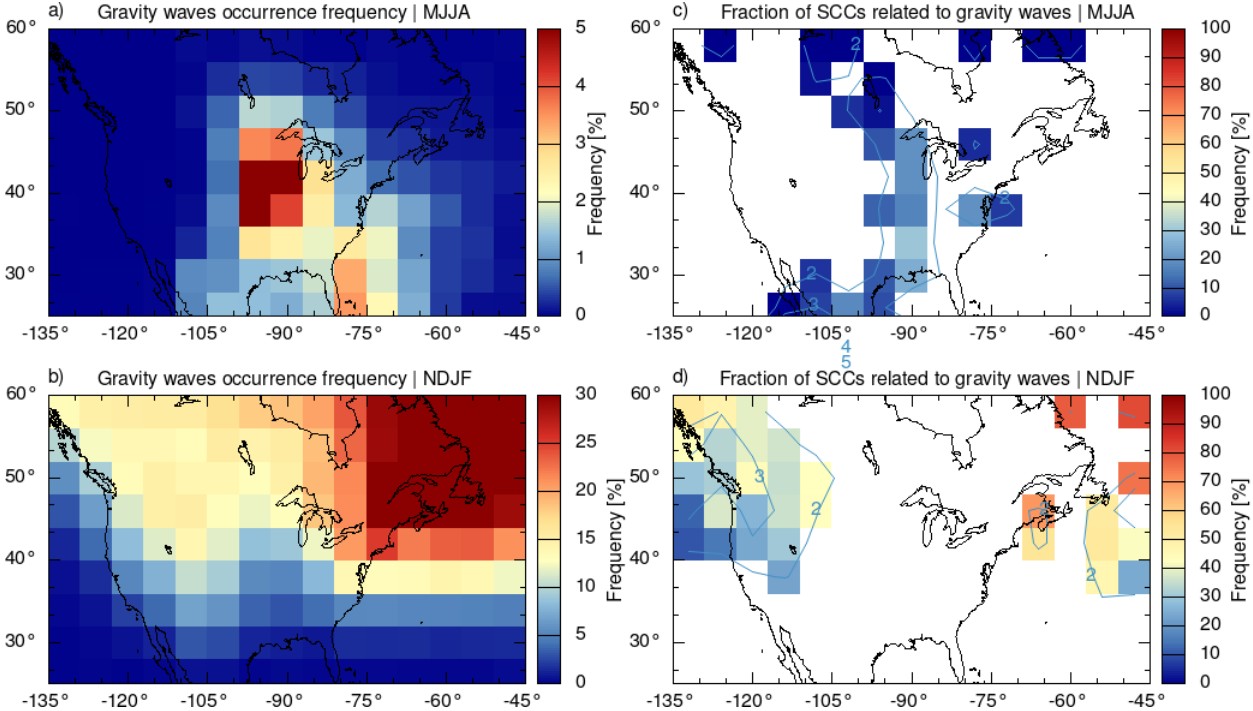

**Figure 12.** Mean occurrence frequency of gravity waves derived from AIRS measurements between 2006 and 2018 in MJJA (a) and NDJF (b). Fraction of SCCs related to gravity waves in MJJA (c) and NDJF (d), here SCCs are those indirectly related to deep convection as found in Fig. 8.The blue contour lines show the occurrence frequencies of SCCs.

the maximum is located over the Great Plains (∼5 %) (Fig. 12a). Frequencies and locations of gravity waves found here are
consistent with results of Hoffmann and Alexander (2010), Hoffmann et al. (2013) and de Groot-Hedlin et al. (2017). In NDJF, gravity waves may explain as much as 50 % of the remaining SCC observations over the North-Eastern Pacific and western Canada and 90 % over eastern Canada and the north-west Atlantic (Fig. 12d). In MJJA, the influence of gravity waves on occurrence of SCCs is weaker ( maximally 30 % over the Great Plains, Fig. 12c), as deep convection plays a more important role.

**4  Discussion**

The result that up to 74 % of SCCs in MJJA over the Great Plains and about 50 % in NDJF over the western Atlantic Ocean, the North-Eastern Pacific, and western Canada are correlated with deep convection implies that deep convection is the major source for SCCs in these regions for the respective seasons. This finding appears plausible, as for tropical SCCs, there is a clear relation with the seasonally varying deep convection (Pan and Munchak, 2011). During the thunderstorm season in MJJA, the
region over the Great Plains is known to produce extraordinarily strong convection at midlatitudes. Their tropopause reaching cloud occurrence frequency is comparable with (daytime)/even higher than in (nighttime) the tropical deep convection hotspots



(Hoffmann et al., 2013). Also in NDJF the occurrence frequencies of convective tropopause reaching clouds over the western Atlantic Ocean, the North-Eastern Pacific, and western Canada is comparable to tropical hotspots (Hoffmann et al., 2013). Moreover, deep convection as a source for SCCs is supported by studies reporting measurements of enhanced water vapor in

the lowermost stratosphere over north America that were attributed to overshooting convection (Barrett et al., 1950; Hanisco et al., 2007; Herman et al., 2017). For the extra-tropics, Spang et al. (2015) identified quasi-isentropic transport of water vapour rich air from the subtropical upper troposphere into the lowermost extra-tropical stratosphere as a source for SCCs based on two selected days out of eight days of CRISTA2 measurements in August 1997. Their findings were more for SCCs located over Scandinavia and the Baltic Sea, both not known for strong convection and out of the region of our investigation. Given

the different regions, times, and time frames investigated we assume that at midlatitudes the dominating SCCs formation mechanisms may also depend on the longitude, orography, land-sea distribution or temperature contrasts.

In this study, we found ∼10% of all deep convection events were correlated with SCCs (Fig. 7). As CALIPSO underestimates the occurrence frequencies of SCCs by a factor of 2 at midlatitudes (Zou et al., 2020), there would be probably even more deep convection events correlated with SCCs. As convection injections reach up to $2-4$ km above the tropopause (Homeyer et al.,

2014a), deep convection is a vital pathway for transporting trace gases, such as e.g. water vapor, and other pollutants into the lowermost stratosphere. Cooney et al. (2018) and Solomon et al. (2016) found the tropopause penetrating convection is common over the central United States in later-spring and summertime, and overshooting events are averagely about 8 % and 12 % at 0000 UTC in monthly all events according to NEXRAD network. Deep convection would play a significant role in regulating the water vapor in the lower stratosphere and affecting the stratosphere and troposphere exchange over North

America, especially in summer season.

The occurrence of SCCs is also affected by the transport from deep convection. We found that fractions of SCCs due to deep convection increase by ∼10 % over North America with a larger search radius (in $6° \times 8°$ at latitude × longitude) for deep convection (Fig. 10 compared with Fig. 6). However it is not entirely clear if the SCCs in the example above (Fig. 9) observed more than 12 hours and several hundred kilometers away from the deep convection source were directly injected or if the

SCCs formed in situ from injected water vapour. Nearly the same SCC occurrence frequencies in MIPAS day- and nighttime measurements (∼10:30 local equator overpass time) (Zou et al., 2020) suggest that SCCs may have a life time longer than 12 hours and exceeding those of the convective cells. While a microphysical simulation study shows that the conditions at the midlatitude tropopause limit the cirrus cloud life time to less than 9 hours due to fast sedimentation (Jensen et al., 1996), and hence suggest that these SCCs away from the deep convection centres may have formed later. The life time of SCCs needs to

be further investigated.

As more than 95 % of the gravity waves over the Great Plains (36 to 46°N, 98 to 88°W) are triggered by deep convection in MJJA (Hoffmann and Alexander, 2010), it is difficult to clearly disentangle the relations of gravity waves and deep convection with the SCCs occurrences. In this work, we counted only SCCs related to gravity waves in absence of deep convection in the same grid box resulting in up to 50 % of the SCC observations over the North-Eastern Pacific and western Canada in NDJF and

30 % over the Great Plains in MJJA. The fraction of SCCs related to gravity waves would be higher (up to $60-70$ % over the Great Plain in MJJA and 90 % over the North-Eastern Pacific and western Canada in NDJF) if data above convective systems





were included. As gravity waves propagate away from the centre of convection and induce temperature perturbations, they might play a role in the formation and maintenance of the SCCs as they have an impact on the particle size distribution by favouring the formation of more but smaller ice particles that consequently sediment slower (Haag and Kärcher, 2004). The

gravity wave breaking at overshooting deep convection cloud tops induces strong upward motion in the lower stratosphere (Qu et al., 2020) that may counteract the sedimentation of the SCC ice particles (Podglajen et al., 2018). The relationship between SCCs and gravity waves is complex and intricate and requires further investigation.

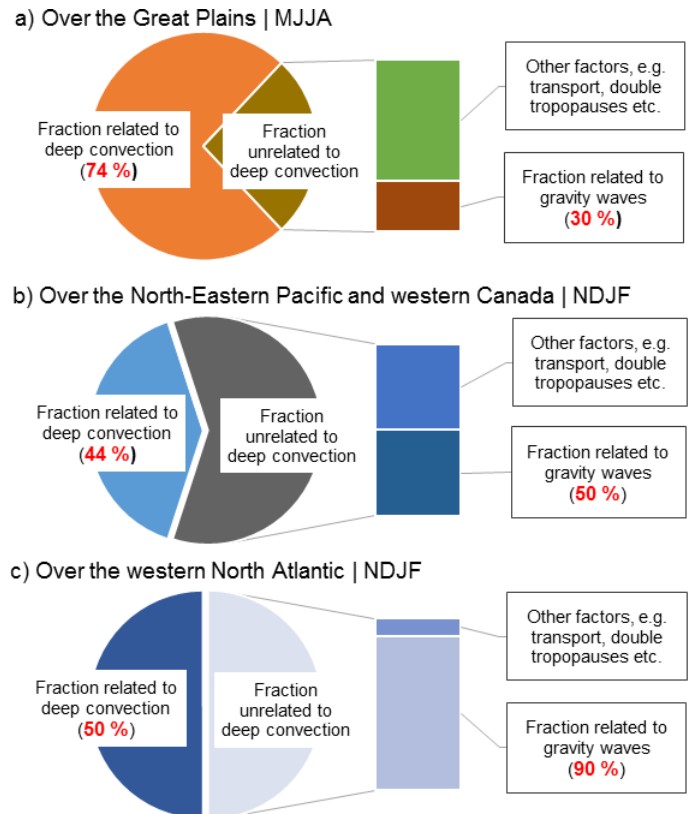

**Figure 13.** Maximum contribution fraction of deep convection, gravity waves and other factors related to the occurrence of SCCs over the Great Plains in MJJA (a), over Nother-Eastern Pacific and western Canada in NDJF (b) and over western North Atlantic in NDJF (c). Note that the maximum occurrence frequencies of SCCs are about 6 % over the Great Plains in MJJA, 4.4 % over North-Eastern Pacific and western Canada and 5.5 % over the western North Atlantic in NDJF.

The mechanisms associated with SCC observations are manifold. Figure 13 summarizes our findings regarding the maximum fraction of SCCs related to deep convection and gravity waves in the summer season over the Great Plains and winter season

over North-Eastern Pacific and western Canada and western North Atlantic. The occurrence of SCCs is mainly associated with deep convection, which coincides with nearly half of the CALIOP SCC observations over winter season hotspots and more than 70 % over summer season hotspots. For those SCCs unrelated to deep convection we identified a strong relation





with gravity waves over the North-Eastern Pacific and western Canada, and the western North Atlantic in NDJF. The fraction correlated with gravity waves is relatively small over the Great Plains in MJJA reaching up to 30 %. The remaining (a) 18 %,

b) 28 %, c) 5 %) SCC observations may be related to quasi-isentropic transport (Spang et al., 2015), double tropopauses (Noël and Haeffelin, 2007; Homeyer et al., 2014b) or baroclinic instabilities connected with upward transport via warm conveyor belts (WCB) (Eckhardt et al., 2004), but these were not investigated in detail here.

## 5  Conclusions

Cirrus clouds in the lowermost stratosphere are of importance for understanding the water vapor budget and the chemical

composition of the stratosphere and their long-term influences on the radiation budget of the Earth's atmosphere will potentially affect climate change. In this work, we quantified the occurrence frequency of SCCs over North America from 13 years (2006 to 2018) of CALIPSO measurements and studied the relations of SCCs with deep convection and stratospheric gravity wave observations from AIRS measurements.

Hotspots of SCCs are observed at nighttime over the North American Great Plains in MJJA with a maximum occurrence

frequency of up to 6.2 % and over the North-Eastern Pacific, western Canada and the western North Atlantic in NDJF with frequencies of up to 5.5 % and 4.4 % . The deep convection observations from AIRS show similar hotspots as the SCCs, even in individual years. The seasonal distributions and daily time series of SCCs and deep convection show a high degree of relation for the Great Plains hotspot. By analyzing the location and time of the SCC and deep convection observations, we found that up to 74 % of SCCs can be attributed to deep convection over the Great Plains in MJJA and about 50 % over the western Atlantic

Ocean, the North-Eastern Pacific and western Canada in NDJF (Fig. 6).

After filtering the SCC events related to local deep convection, there still remain up to 3 % of SCC observations over the Great Plains in MJJA and over the North-Eastern Pacific, western Canada and the western North Atlantic in NDJF. An example of backward trajectories from SCC observations calculated with a Lagrangian model particularly points out the role of transport processes. Following this, we found the contribution of deep convection to SCC occurence to have increased by ∼10 % when

enlarging the search radius for deep convection.

Other processes can be related to SCC observations as well. By correlating the CALIOP SCC observations with AIRS observations of stratospheric gravity waves, we found that about 30 % of the SCCs events that are unrelated to deep convection over the Great Plains in MJJA are correlated with gravity wave observations. In NDJF the fractions are even larger, i. e., 50 % over the North-Eastern Pacific and western Canada as well as up to 90 % over eastern Canada and the north-west Atlantic.

Gravity waves also show close relation with SCCs over North America. However, due to the high relation between gravity waves and deep convection, it's difficult to separate the contributions of them precisely.

In this study, we analyzed the distribution of cirrus clouds in the lowermost stratosphere and the meteorological phenomena associated with SCCs over North America. We found that, locally and seasonally, deep convection is the leading factor related to the occurrence of SCCs over North America. Gravity waves and transport from deep convective sources also have high

relations with SCCs. In this work we focused on the region of North America, where deep convection takes a dominating role





in affecting the SCC presence and distribution. For a wider and deeper knowledge of the formation mechanism and distribution of SCCs, we will look at global observations of SCCs and try to identify the relevant sources and processes in future work.

*Data availability.* Convection and gravity waves data from AIRS are available at https://www.re3data.org/repository/r3d100012430 (last access: 3 December 2020). ERA-Interim tropopause data are available at https://www.re3data.org/repository/r3d100013201 (last access: 25 May 2020). Cirrus cloud top heights from CALIPSO are available upon request from the contact author, Ling Zou (l.zou@fz-juelich.de; cheryl_zou@whu.edu.cn).

## Appendix A: Sensitivity test regarding the detection of deep convection

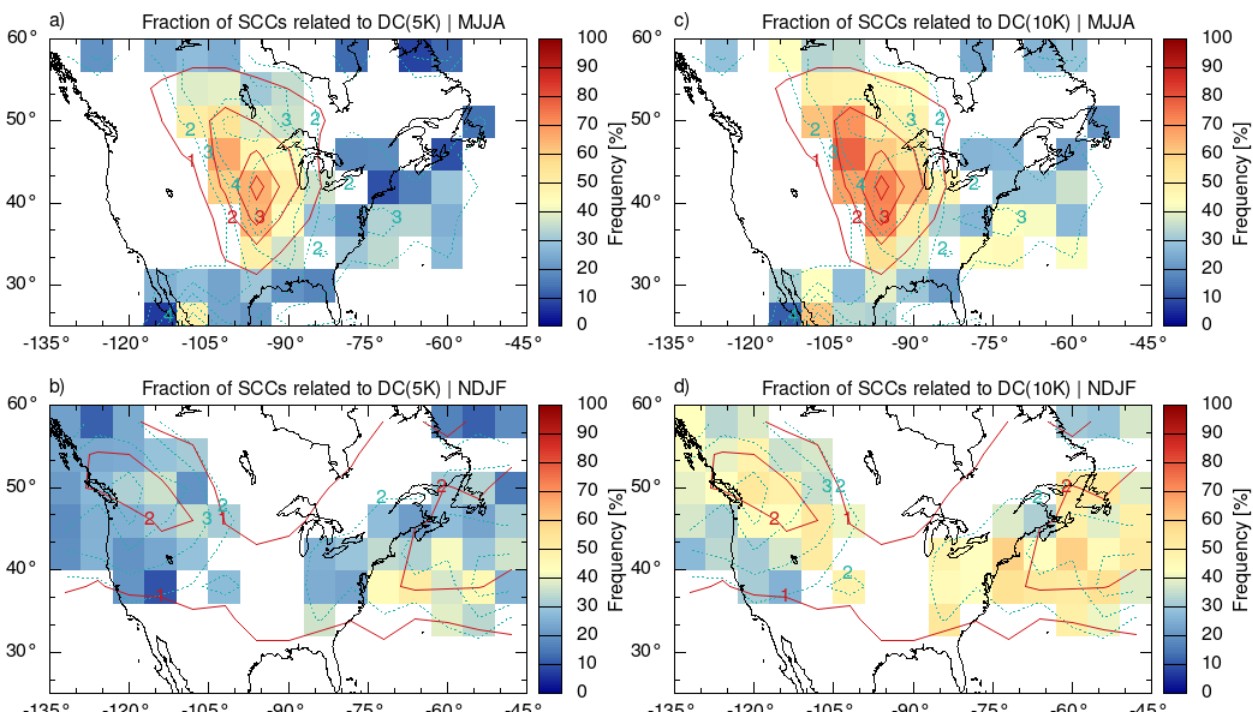

**Figure A1.** Fraction of SCCs related to deep convection with temperature threshold of 5 K (a and b) and 10 K (c and d), red contour lines are the occurrence frequency of deep convection and blue dotted contour lines are the occurrence frequency of SCC.

A brightness temperature threshold of 7 K with respect to the tropopause temperature was used to identify deep convection events or mesoscale convective systems from AIRS observations in this work. By changing the threshold to 5 K or 10 K, we will obtain different values of deep convection occurrence frequency and corresponding fractions of SCCs related to deep convection (Fig. A1). It should be considered that smaller temperature thresholds (5 K) may miss overshooting convection and larger thresholds (10 K) will include some convection events that are restricted to the troposphere. The maximum fractions of



SCCs related to deep convection in MJJA over the Great Plains are about 65 % and 80 % with temperature threshold of 5 K and 10 K, respectively. 7 K is the threshold we finally selected based on the vertical uncertainties of ERA-Interim vertical resolution

and tropopause heights. No matter the temperature threshold is larger or smaller than 7 K, the conclusion that deep convection is the primary force related to the occurrence of SCCs over North America is rather robust.

*Author contributions.* LZ, LH, and SG conceived the study design. LH provided the AIRS data and the ERA-Interim/ERA5 tropopause data. LZ processed the CALIPSO data and compiled all results. RS provided useful discussion and valuable comments on manuscript. LCW provided helpful discussion about the CALIPSO data processing. LZ wrote the manuscript with contributions from LH and SG.

*Competing interests.* The authors declare that they have no conflict of interest.

*Acknowledgements.* Dr. Ling Zou was supported by the National Natural Science Foundation of China under grant No. 41801021 and the International Postdoctoral Exchange Fellowship Program 2018 under grant No. 20181010. This work was supported by the German Research Foundation (DFG) through the AeroTrac project under the grant ID: DFG HO5102/1-1. We thank Dr. Olaf Stein from the Forschungszentrum Jülich for useful feedback. We gratefully acknowledge the computing time granted on the supercomputers JURECA and JUWELS at

Forschungszentrum Jülich. CALIPSO data are obtained from the NASA Langley Research Center Atmospheric Science Data Center. The AIRS data were distributed by the NASA Goddard Earth Sciences Data Information and Services Center. The ERA-Interim reanalysis data were obtained from the European Centre for Medium-Range Weather Forecasts.



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
