# Peer review of "Empirical evidence for deep convection related stratospheric cirrus clouds over North America"

_Atmospheric Chemistry and Physics, 2021_

## Community Comment (CC2)

Dear Dr. Jie Gong,

Thank you very much for your interest in our study. Please find our replies to your comments below.

**Comment ∗ 1**

I think there's a fundamental issue with identification of SCC from CALIPSO level-1 532 nm backscatter. First, from the example given in Fig. 1, this is an convection overshooting, instead of a thin cirrus cloud.

**Answer:**

In this work, we intended to study lower stratospheric ice clouds, including all ice clouds with the cloud top being located above the tropopause. We didn't distinguish between detections of cirrus, cirrostratus or deep convective clouds. Tropopause penetrating convective clouds as shown in Fig. 1 are included in our study, as deep convection is a major cause for the occurrence of ice in the midlatitude lower-most stratosphere. We would clarify this in revised manuscript.

**Comment ∗ 2**

Secondly, stratospheric feature could be aerosol. Why not use Level-2 cloud type from Version 4.X feature classification flag?

**Answer:**

In this study, we used the Level 2 Vertical Feature Mask data from CALIOP Version 4.X and not the backscatter data to detect the stratospheric ice clouds. The 532 nm backscatter signal in Figure 1 is shown only for illustration. However, we found it would be helpful to also show the Vertical Feature Mask data for this example in the revised manuscript, please see revised Figure 1 below.

**Comment ∗ 3**

I really doubt if there's many stratospheric thin cirrus, as there's no immediate mechanism there other the reminiscence of overshooting cloud top. See Avery et al. (2017, https://www.nature.com/articles/ngeo2961) for the overshooting cloud.

**Answer:** As answered in comment 1, we investigated actually not only thin cirrus clouds but all ice clouds in the lower stratosphere.

**Comment ∗ 4**

Thirdly, as tropopause height has 600-1000 m uncertainty in ERA-Interim, I think using 500m as a threshold to determine the separation between troposphere and stratosphere is dangerous. Besides, tropopause is a layer, not a thin interface. This is a minor issue compared with the first and second issue.

**Answer:**

We are aware that ERA-Interim data have about 600-1000 m vertical resolution in the UT/LS region. Nevertheless, we consider a threshold of 500 m with respect to the tropopause height derived from ERA-Interim to be useful to detect stratospheric ice clouds for the following reasons:

Firstly, the ERA-Interim data have been interpolated to a much finer vertical grid using cubic spline interpolation in order to improve the vertical resolution of the geopotential height estimates of the tropopause. This method has already been applied in Spang et al. (2015) to produce an ERA-Interim-based 'high-resolution' tropopause height data set.

Secondly, Tegtmeier et al. (2020) found lapse rate tropopause (LRT) height differences between ERA-Interim and Global Navigation Satellite System-Radio Occultation (GNSS-RO) observations of about 200 m in the tropics (Figure 8 in Tegtmeier et al. (2020), Figure 2 shown below). This shows that

uncertainties in ERA-Interim tropopause heights can be significantly smaller than the vertical resolution of the data. So, the 500 m threshold for ERA-Interim data is reasonable.

Thirdly, 500 m is comparable to thresholds used to detect stratospheric ice clouds in previous studies, e. g. Homeyer et al. (2010) and Pan and Munchak (2011).

Therefore, we consider 500 m as a valid threshold with respect to the tropopause derived from ERA-Interim to detect lower stratosphere ice clouds. We will explain the uncertainties of the ERA-Interim tropopause data more in the revised manuscript.

**References**

Homeyer, C. R., Bowman, K. P., and Pan, L. L.: Extratropical tropopause transition layer characteristics from high-resolution sounding data, Journal of Geophysical Research Atmospheres, 115, D13 108, https://doi.org/10.1029/2009JD013664, 2010.

Pan, L. L. and Munchak, L. A.: Relationship of cloud top to the tropopause and jet structure from CALIPSO data, Journal of Geophysical Research Atmospheres, 116, 1–17, https://doi.org/10.1029/2010JD015462, 2011.

Spang, R., Günther, G., Riese, M., Hoffmann, L., Müller, R., and Griessbach, S.: Satellite observations of cirrus clouds in the Northern Hemisphere lowermost stratosphere, Atmospheric Chemistry and Physics, 15, 927–950, https://doi.org/10.5194/acp-15-927-2015, 2015.

Tegtmeier, S., Anstey, J., Davis, S., Dragani, R., Harada, Y., Ivanciu, I., Pilch Kedzierski, R., Krüger, K., Legras, B., Long, C., Wang, J. S., Wargan, K., and Wright, J. S.: Temperature and tropopause characteristics from reanalyses data in the tropical tropopause layer, Atmospheric Chemistry and Physics, 20, 753–770, https://doi.org/10.5194/acp-20-753-2020, 2020.

[Figure]

Figure 1: Lower stratosphere ice clouds observed above deep convection on 12 June 2018. a) 8.1 $\mu$m brightness temperatures (BT) derived from AIRS overlayed with CALIPSO orbits (black lines). The red circles along the CALIPSO orbits indicate ice clouds with cloud top heights being at least 500 m above the tropopause. b) 532 nm total attenuated backscatter (black dashed line is ERA-Interim LRT) and vertical feature mask data for the clouds of interest.

[Figure]

**Figure 8.** Latitudinal distributions of zonal-mean lapse rate tropopause temperature (**a**), altitude (**b**), and pressure (**c**) based on radio occultation data and reanalysis products during 2002–2010 (**a–c**). Differences between radio occultation and reanalysis estimates are shown in the panels (**d**)–(**f**).

Figure 2: Figure from Tegtmeier et al. (2020)

---

## Author Comment (AC1)

Dear Reviewer,

Thank you very much for your time and efforts on reviewing our manuscript. We considered all comments and suggestions, and provided our detailed point-by-point responses below.

**Reviewer #1**

**General Comments**

This is an interesting paper examining the frequency and distribution of clouds in the lowermost stratosphere over North America from many years of satellite-based lidar observations. It is generally demonstrated that stratospheric clouds are found commonly over the central Great Plains of the United states during the warm season and over the northwestern portion of North America and Atlantic Ocean during the cool season. It is further argued by comparing with satellite-based classification of deep convection that most of the stratospheric clouds are closely located (or juxtaposed) with convection. Much of the remaining clouds identified are linked with gravity waves (which are in large initiated by deep convection). These linkages are made somewhat loosely, as is common in a lot of prior work, but the association is logical and mostly appropriate. Some attempts to deal with spatial offsets between ongoing deep convection and stratospheric clouds are carried out and shown to have a minor, but important, effect on the results. Examining the frequency of stratospheric clouds is important and justified well in the paper, but some efforts to strengthen the evaluation and messaging throughout are recommended below.

**Answer:** Thank you very much.

**Comment ∗ 1**

On the characterization of stratospheric clouds as "cirrus": based on my evaluation of the paper, I expect a great deal of the stratospheric clouds identified are indeed overshooting cumulonimbus clouds rather than distinct (or mostly separate) cirrus clouds. To avoid confusion or misinterpretation of the focus of this work, I would recommend that the authors refer to clouds examined here as stratospheric clouds (or stratospheric ice clouds).

**Answer:** Thank you for your suggestion. The term "stratospheric ice clouds (SICs)" would be more appropriate in this work. We have revised them all in our revised manuscript.

**Comment ∗ 2**

One theme of the paper that I think needs a bit of work is the messaging throughout on the potential sources / formation mechanisms for the stratopheric clouds examined. Deep convection is discussed as only delivering water vapor to the stratosphere (e.g., see lines 74-86; 195-196; 328-330 as examples), when in fact many of the studies the authors cite demonstrate that the primary pathway for delivering water to the stratosphere is via ice. Thus, much of the clouds examined (especially those linked to deep convection) are very likely residual cloud material from previous injection. The stratosphere is often far too dry otherwise to enable in situ cirrus formation, for which some requested additions to the paper below may help sort out.

**Answer:** Yes, both water vapor and ice particles would possibly be injected into lower stratosphere by deep convection. We revised those related sentences in the revised manuscript. Please also find the revisions below.

"Considering the pathways of air into the midlatitude stratosphere, deep convection, especially tropopause-penetrating convection, could produce direct injection of water vapor and ice particles into

the middleworld and even the overworld stratosphere (Weinstock et al., 2007)."

"As convection injections reach to $2-4\,km$ above the tropopause (Homeyer et al., 2014a), deep convection is a vital pathway for transporting ice particles, water vapor, and other pollutants into the lowermost stratosphere."

**Comment $*$ 3**

There is some analysis given to demonstrate sensitivity of linkages between convection and stratospheric clouds to the thresholds used for deep convection classification, which is helpful. For broader context of the occurrence of stratospheric clouds and their potential impact, I would recommend the authors add a figure to show the tropopause-relative altitude distribution of stratospheric clouds identified (by season). It would also be helpful to know what these distributions looked like for populations linked with deep convection and gravity waves. Because the background stratospheric water vapor concentration is 3-6 ppmv (with higher concentrations sometimes found within 1 km of the tropopause), I would expect the gravity wave process to be largely confined to the lowest kilometer of the stratosphere. This would also help to provide confidence in some of your linkages and their seasonality and allow you to assess potential sensitivity of your results to the tropopause-relative altitude threshold used.

**Answer:** We added the occurrence frequency of ice cloud at different altitude ranges with respect to tropopause in Figure 2 and Figure 7.

"The vertical occurrence frequencies of ice cloud top heights with respect to the tropopause in Fig. 2c and Fig. 2d show that, most ice cloud top heights over North America are observed around the tropopause $(7-8\,\%)$. The occurrence frequencies of SIC over North America are about $2\,\%$ both in summer and winter."

"Vertically, the highest fractions of ice cloud top heights unrelated to deep convection are also located at tropopause region (Fig. 7c, d). The fractions of SIC are about $1\,\%$ over North America, which decrease by $50\,\%$ compared to Fig. 2."

[Figure]

Figure 2: Nighttime seasonal mean occurrence frequencies of SICs from 2006 to 2018 in a) MJJA and b) NDJF on a 4° × 6° (latitude × longitude) grid size. detected by CALIPSO are cirrus clouds that have a cloud top height being at least 500 m above the tropopause derived from ERA-Interim. c) and d) are the corresponding occurrence frequencies of ice cloud top heights in the altitude range from −4 to 4 km with respect to the tropopause over North America. Occurrence frequencies of deep convection are shown in e) MJJA and f) NDJF. Deep convection events measured by AIRS are defined as a difference of the 8.1 μm brightness temperature and the ERA-Interim tropopause temperature below 7 K. The red contour lines in a) and b) show the occurrence frequencies of deep convection and blue contour lines in e) and f) are occurrence frequencies of SICs.

[Figure]

Figure 7: Occurrence frequencies of SIC observations that are not directly related to deep convection for MJJA (a) and NDJF (b). The red contour lines show the occurrence frequencies of deep convection and blue dotted contour lines are the occurrence frequencies of SICs. c) and d) are the fractions of ice cloud top heights relative to the tropopause, which are not directly related to deep convection.

**Comment ∗ 4**

The potential significance of poleward transport of water from the TTL to aid in stratospheric cloud formation is a bit overstated in my opinion. While there has been some evidence that this can occur, much of the broader analyses conducted (especially those looking at double-tropopause versus single-tropopause regions; e.g., see Schwartz et al 2015 - http://doi.org/10.1002/2014JD021964) have demonstrated that such transport is typically far drier than the extratropical lower stratosphere. Thus, I expect such a stratospheric cloud source to be exceedingly rare. I recommend the authors expand the discussion to point out this limitation.

**Answer:** As the poleward transport of water vapor from TTL to midlatitudes is presented in both Spang et al. (2015) and Schwartz et al. (2015), we considered it as a potential mechanism for the formation of cirrus at midlatitudes, which are not explained by deep convection and gravity waves. But yes, the moistening effect of this transport is limited (Schwartz et al., 2015). We have rephrased related sentences in the revised manuscript. But as the fractions of isentropic transport and double tropopause on the formation of ice clouds is uneasily to be quantified, we are not going to discuss them in detail in this study.

"Moreover, other factors may contribute to the occurrence of high altitude ice clouds, such as, poleward transport of water vapor from the TTL to the extratropics(Dessler et al., 1995; Pittman et al., 2007)."

**Comment ∗ 5**

Much of the remaining uncertainty in the association of stratospheric clouds with convection comes from the lack of an exploration of time offsets between convection occurrence and stratospheric cloud detection. This is explained somewhat generally as "atmospheric transport" in the paper, but I believe it should be given more attention/discussion. I suspect much of the residual could be linked to convection with more trajectory analysis and consideration of time offsets between storm occurrence and stratospheric cloud detection. The lifetimes of clouds in the stratosphere following convective injection (as summarized in the paper) and the downstream offsets of distributions shown certainly support the argument that much of those clouds that do not directly coincide with convection are linked closely in time with prior tropopause-reaching/overshooting convection. I am not necessarily suggesting the authors conduct such trajectory analyses, but they could do so for the instances where they don't have a clear linkage between stratospheric clouds and deep convection/gravity waves as they already demonstrated this conceptually in Figure 9 and it would be a nice addition to the paper. Improving the discussion throughout to emphasize a potential timing offset would help clarify the sources of uncertainty mentioned and contrast well with the spatial offsets which are given a fair amount of attention.

**Answer:** Thanks for the suggestion. Trajectory analyses are very interesting to investigate the lifetime of ice clouds and the time offsets between convection and ice cloud occurrence. We would conduct this analysis in our future work as this task is complex and computationally expensive for 13 years of input data. However, we added two sentences to the discussion of the revised manuscript.

"Further investigation of the lifetime of SICs and the temporal offsets between the occurrence of convection and the detection of SICs by means of Lagrangian trajectory models would help to identify the sources of some unexplained SICs. As the trajectory analyses with large input data would be complex and rather computationally expensive, we foresee it for future work."

**Comment ∗ 6**

The summary of past ground-based (and some satellite-based) analyses of overshooting convection over North America is good, but there are several recent GPM studies of overshooting that should be listed here and would help round out the discussion: Liu Liu (2016) - http://doi.org/10.1002/2015JD024430, and Liu et al. 2020 - https://doi.org/10.1029/2019JD032003.

**Answer:** Thanks for those helpful references. They were included in the revised manuscript.

"Based on one year (March 2014 to February 2015) (Liu and Liu, 2016) and 4 years (April 2014 to March 2018) (Liu et al., 2020) of Global Precipitation Measurement (GPM) Precipitation Feature (PF) data, larger area and higher occurrence of 20 dBZ radar reflectivity at and above tropopause were observed in northern middle and high latitudes than over tropics. As the 20 dBZ radar reflectivity above the tropopause is defined as overshooting convection in Liu and Liu (2016) and Liu et al. (2020), the high occurrence of it means lots of air and ice particles were directly injected above the tropopause in midlatitudes."

**Specific Comments / Technical Corrections**

Lines 37-38: "two-third" should be "two-thirds", and "actually was" should be "is"
Lines 59-60: What is meant by "kind of controversial" here? It doesn't follow the argumentation leading this, so please be more explicit.
Line 62: "direct injection of air" and ice!
Line 67: "central" should be "central Great Plains"
Line 82: "up to 1-6 km" should be "up to 6 km". When specifying a maximum possible altitude, specifying ranges should be avoided.
Line 83: "convective system" should be "convective systems"
Line 90: "lead" should be "leads"
Line 133: "cirrus clouds top" should be "cirrus cloud top"
Line 177: "" should be "m"
Lines 238-240: I would recommend also pointing out duration!
Figure 4: caption says "purple lines", but these appear blue to me.
Figure 5: why is the green line in the top panel a different color here?
Line 319: "up to 1-4 km" should be "up to 4 km", though note earlier it was stated that this could be up to 6 km (see line 82 comment above).
Line 322: "are averagely about". What exactly do you mean here? Please clarify.

**Answer:** Done in the revised manuscript. And please also find them below.

1. However, as shown by Dessler et al. (2013), two-thirds of the stratospheric water vapor feedback actually was from water vapor increasing in the stratosphere in extratropics.

2. Even though the statistical numbers of occurrence frequency are kind of controversial from study to study, the occurrence of SICs at midlatitudes is notable.

3. Considering the pathways of air into the midlatitude stratosphere, deep convection, especially tropopause-penetrating convection, could produce direct injection of water vapor and ice particles into the middle-world and even the overworld stratosphere (Weinstock et al., 2007).

4. Cooney et al. (2018) and Solomon et al. (2016) found that the overshooting events are most common in the north-central part of the United States.

5, 6. Homeyer et al. (2017) found gravity wave breaking was the primary source for above-anvil cirrus plumes, which could reach to $1-6\,\mathrm{km}$ above the tropopause in extratropical convective systems.

7. it turned out that poleward isentropic transport from the upper tropical troposphere to the extratropical stratosphere leads to the occurrence of SICs in these observations.

8. To reliably estimate ice cloud top heights, only high-feature-type-quality cirrus cloud and deep convection data from the CALIPSO V4.x Level-2 Vertical Feature Mask data product are used in this study.

9. Gravity wave events are detected based on a threshold of $0.05\,\mathrm{K}^2$ for the noise-corrected $4.3\,\mu\mathrm{m}$ brightness temperature variances.

10. Note that the linear correlation coefficients of SICs and deep convection wouldn't well represent their relations as the rate at which SICs are formed by deep convection will depend on several parameters such as the intensity, the spatial extent and the duration at which deep convection occurs.

11,12. Please find new figure and revised texts below.

13. As convection injections reach to $2-4\,\mathrm{km}$ above the tropopause (Homeyer et al., 2014), deep convection is a vital pathway for transporting ice particles, water vapor and other pollutants into the lowermost stratosphere.

14. Cooney et al. (2018) and Solomon et al. (2016) found the tropopause penetrating convection is common over the central United States in later-spring and summertime, for example, at 0000 UTC, the average fraction of total monthly overshoots is about $8\,\%$ (Cooney et al., 2018) and the total annual overshooting fraction is about $12\,\%$ (Solomon et al., 2016) based on NEXRAD network.

Answer to Point 11 and 12:

"The temporal correlations between SCCs and deep convection are further analyzed based on time-series of daily detection numbers ($n_{obs}$) over the Midwest United States ($35°\mathrm{N}-45°\mathrm{N}$, $90°\mathrm{W}-100°\mathrm{W}$) from May to August (Fig. 4). Data from three years, 2010, 2013, 2015, which have Pearson linear correlation coefficients ($r_{SIC-DC}$) of 0.66, 0.52, and 0.3, respectively, between the number of CALIPSO SIC observations and AIRS deep convection observations, are shown here as representative examples to illustrate the temporal correlation between SICs and deep convection. A visual inspection of the time series in Fig. 4 shows that individual events or episodes of SICs and deep convection often occur simultaneously.

However, as the rate at which SICs are formed depends on several parameters, such as the intensity, the spatial extent, and the duration of the deep convection events, the simple linear correlation coefficient of the $n_{obs}$ of SICs and deep convection is not necessarily the best indicator for correlation. Additionally, we considered the number of days $NOD$ with SICs and deep convection detections as a proxy for identifying possible correlations. In Fig. 4, we see that even in 2015, which has the lowest linear correlation coefficient, $87\,\%$ of the days with SIC observations are related to days with deep convection. SICs are generally co-occurring with deep convection ($>80\,\%$ for each single year from 2007 to 2018, not shown), which indicates a high degree of correlation between deep convection and SICs on the temporal scale over the Great Plains in summertime."

[Figure]

Figure 4: Daily numbers of nighttime observations of SICs (red line with no offset), deep convection (blue line with offset of 300) over the Midwest United States ($35°N-45°N$, $90°W-100°W$) in MJJA in 2010, 2013 and 2015. Number of days with occurrences of SICs ($NOD_{SIC}$), deep convection ($NOD_{DC}$) and both of them ($NOD_{both}$) are counted. Total detection numbers of SICs ($n_{SIC}$), deep convection ($n_{DC}$) on each day and their Pearson linear correlation coefficients ($r_{SIC-DC}$) are also shown above the plots.

**References**

Cooney, J. W., Bowman, K. P., Homeyer, C. R., and Fenske, T. M.: Ten Year Analysis of Tropopause-Overshooting Convection Using GridRad Data, Journal of Geophysical Research: Atmospheres, 123, 329–343, https://doi.org/10.1002/2017JD027718, 2018.

Dessler, A. E., Schoeberl, M. R., Wang, T., Davis, S. M., and Rosenlof, K. H.: Stratospheric water vapor feedback, Proceedings of the National Academy of Sciences, 110, 18 087–18 091, https://doi.org/10.1073/pnas.1310344110, 2013.

Homeyer, C. R., Pan, L. L., and Barth, M. C.: Transport from convective overshooting of the extratropical tropopause and the role of large-scale lower stratosphere stability, Journal of Geophysical Research: Atmospheres, 119, 2220–2240, https://doi.org/10.1002/2013JD020931, 2014.

Homeyer, C. R., McAuliffe, J. D., and Bedka, K. M.: On the Development of Above-Anvil Cirrus Plumes in Extratropical Convection, Journal of the Atmospheric Sciences, 74, 1617–1633, https://doi.org/10.1175/JAS-D-16-0269.1, 2017.

Liu, N. and Liu, C.: Global distribution of deep convection reaching tropopause in 1year GPM observations, Journal of Geophysical Research: Atmospheres, 121, 3824–3842, https://doi.org/10.1002/2015JD024430, 2016.

Liu, N., Liu, C., and Hayden, L.: Climatology and Detection of Overshooting Convection From 4 Years of GPM Precipitation Radar and Passive Microwave Observations, Journal of Geophysical Research: Atmospheres, 125, e2019JD032 003, https://doi.org/10.1029/2019JD032003, 2020.

Schwartz, M. J., Manney, G. L., Hegglin, M. I., Livesey, N. J., Santee, M. L., and Daffer, W. H.: Climatology and variability of trace gases in extratropical double-tropopause regions from MLS, HIRDLS, and ACE-FTS measurements, Journal of Geophysical Research: Atmospheres, 120, 843–867, https://doi.org/10.1002/2014JD021964, 2015.

Solomon, D. L., Bowman, K. P., and Homeyer, C. R.: Tropopause-Penetrating Convection from Three-Dimensional Gridded NEXRAD Data, Journal of Applied Meteorology and Climatology, 55, 465 – 478, https://doi.org/10.1175/JAMC-D-15-0190.1, 2016.

Spang, R., Günther, G., Riese, M., Hoffmann, L., Müller, R., and Griessbach, S.: Satellite observations of cirrus clouds in the Northern Hemisphere lowermost stratosphere, Atmospheric Chemistry and Physics, 15, 927–950, https://doi.org/10.5194/acp-15-927-2015, 2015.

Weinstock, E. M., Pittman, J. V., Sayres, D. S., Smith, J. B., Anderson, J. G., Wofsy, S. C., Xueref, I., Gerbig, C., Daube, B. C., Pfister, L., Richard, E. C., Ridley, B. A., Weinheimer, A. J., Jost, H.-J., Lopez, J. P., Loewenstein, M., and Thompson, T. L.: Quantifying the impact of the North American monsoon and deep midlatitude convection on the subtropical lowermost stratosphere using in situ measurements, Journal of Geophysical Research: Atmospheres, 112, https://doi.org/10.1029/2007JD008554, 2007.

---

## Author Comment (AC2)

Dear Reviewer,

Thank you very much for your time and efforts on reviewing our manuscript. We considered all comments and suggestions, and provided our detailed point-by-point responses below.

**Reviewer #2**

**General Comments**

This well-written article presents a good analysis of stratospheric cirrus clouds above North America, and provides convincing explanations for the mechanisms responsible for the presence of these clouds. The data is well described, the figures support the results and the discussion is interesting. I am in favour of its publication in ACP, once the minor comments below will be addressed.

A semantic issue I'd like to see addressed is the fact that the clouds described as "SCC" in the present article appear to be most often than not, according to the results presented, overshoots of convective systems. Labelling them as "stratospheric cirrus clouds" makes me a bit uneasy, as they are most frequently not independent clouds but rather a small part of a larger system, that happens to reach above the tropopause. Depending on the nature of a stratospheric cloud (independent cirrus or upper part of a convective system), I suppose the formation processes involved should be very different, as should be the impact on stratospheric water vapour. It would be useful if the authors could address this issue, either by proposing a way to avoid confusing independent cirrus with upper parts of convective systems, or by demonstrating that the distinction is not important.

**Answer:** Thank you very much for your positive comments.

Yes, we considered the term "stratospheric ice clouds (SICs)" would be more appropriate in this work as it includes also the stratospheric part of deep convection. We have revised them all in the revised manuscript.

**Minor Comments**

**Comment * 1**

Introduction: this section is very good. The authors summarised very well the existing literature on stratospheric cirrus clouds, the observational evidence for their existence, and the mechanisms that might lead to their formation. It is an enjoyable read. L. 59: It is unclear to me what you mean by "kind of controversial". Are you suggesting the results from the previously cited works are incorrect, inconclusive, or unreliable? Maybe you mean that existing evidence for stratospheric cirrus is circumstantial and imprecise by definition (there are multiple definitions of the tropopause, etc). If that is the case, the presented results could also be labelled controversial. Please be more explicit in your statement.

**Answer:** In this sentence, we intend to mention that even though the specific statistics of occurrence frequencies of ice clouds in midlatitudes are different from study to study, the existence of those high altitude ice clouds is well accepted. We have rephrased it in the revised manuscript.

"Even though the actual numbers of occurrence frequency are different from study to study, the occurrence of SICs at midlatitudes is generally notable"

**Comment ∗ 2**

L. 192: "The example adds a further aspect... over North America" I'm not sure I understand this sentence.

**Answer:** We have revised it in manuscript.

"Figure.1 shows a typical example of co-located stratospheric ice clouds and deep convection, a situation which can be frequently observed over the Great Plains during the thunderstorm season."

**Comment ∗ 3**

Figure 4: A visual inspection of the time series presented here does not suggest to me a good correlation between them. Spikes in $N_{SCC}$ (orange) are frequently paired with flat $N_{DC}$ or $BT_{min}$ curves, and vice versa. The text does not attempt to discuss short-scale variability of these time series or of the correlation between the time series. The text does not really discuss the contents of that figure. Because of this, I do not think this figure really brings anything to support the paper's argument. I would actually be more interested in a visual representation of the annual evolution of the various indicators, $NOD_{SCC}$, $NOD_{DC}$, $N_{SCC}$, $R_{SCC\ DC}$, etc. This would make a more interesting discussion in my opinion.

**Comment ∗ 4**

L. 233: "Despite large day-to-day variations, the occurrences of SCC and deep convection are generally correlated." Again, looking at Figure 4, I'm not sure I see such a good correlation. The correlation coefficient between SCC and convection is sometimes as low as 0.3. Do the authors consider a 0.5 correlation coefficient high or low? I would be interested in seeing a lengthier discussion of these parameters, hopefully helped with a figure.

**Answer:** Thanks, we would like to answer comment 3 and 4 together. Figure 4 and corresponding texts have been revised in the revised manuscript.

In Fig.4, we want to show the high correlations between occurrence of stratospheric ice cloud and deep convection. As the rate at which SICs are formed depends on several parameters, such as the intensity, the spatial extent, and the duration of the deep convection events, the simple linear correlation coefficient of the observation numbers of SICs and deep convection is not necessarily the best indicator for correlation. Additionally, we considered the number of days with SICs and deep convection detections as a proxy for identifying possible correlations. And it is the method we used to calculate the fraction of SICs related to deep convection in Section 3.3. Please find details below.

Yes, the annual evolution of deep convection and SICs is very interesting. But the long-term variation analysis seems out of scope of this study, and we will investigate it in our future work.

"The temporal correlations between SCCs and deep convection are further analyzed based on time-series of daily detection numbers ($n_{obs}$) over the Midwest United States (35°N – 45°N, 90°W – 100°W) from May to August (Fig. 4). Data from three years, 2010, 2013, 2015, which have Pearson linear correlation coefficients ($r_{SIC-DC}$) of 0.66, 0.52, and 0.3, respectively, between the number of CALIPSO SIC observations and AIRS deep convection observations, are shown here as representative examples to illustrate the temporal correlation between SICs and deep convection. A visual inspection of the time series in Fig. 4 shows that individual events or episodes of SICs and deep convection often occur simultaneously.

However, as the rate at which SICs are formed depends on several parameters, such as the intensity, the spatial extent, and the duration of the deep convection events, the simple linear correlation coefficient of the $n_{obs}$ of SICs and deep convection is not necessarily the best indicator for correlation. Additionally, we considered the number of days $NOD$ with SICs and deep convection detections as a proxy for identifying possible correlations. In Fig. 4, we see that even in 2015, which has the lowest linear correlation coefficient, 87 % of the days with SIC observations are related to days with deep convection. SICs are

generally co-occurring with deep convection (>80 % for each single year from 2007 to 2018, not shown), which indicates a high degree of correlation between deep convection and SICs on the temporal scale over the Great Plains in summertime."

In Section 3.3:

"The fraction of SICs related to deep convection is defined as the ratio of day numbers with SICs and deep convection both detected to the total number of days with SIC detections."

[Figure]

Figure 4: Daily numbers of nighttime observations of SICs (red line with no offset), deep convection (blue line with offset of 300) over the Midwest United States ($35°N - 45°N$, $90°W - 100°W$) in MJJA in 2010, 2013 and 2015. Number of days with occurrences of SICs ($NOD_{SIC}$), deep convection ($NOD_{DC}$) and both of them ($NOD_{both}$) are counted. Total detection numbers of SICs ($n_{SIC}$), deep convection ($n_{DC}$) on each day and their Pearson linear correlation coefficients ($r_{SIC-DC}$) are also shown above the plots.